# The association of maternal factors with the neonatal microbiota and health

Bin Zhu [1,2], David J. Edwards[2,3], Katherine M. Spaine[1,2], Laahirie Edupuganti[1,2], Andrey Matveyev[1,2], Myrna G. Serrano [1,2] & Gregory A. Buck [1,2,3,4,5] ✉

The human microbiome plays a crucial role in human health. However, the influence of maternal factors on the neonatal microbiota remains obscure. Herein, our observations suggest that the neonatal microbiotas, particularly the buccal microbiota, change rapidly within 24–48 h of birth but begin to stabilize by 48–72 h after parturition. Network analysis clustered over 200 maternal factors into thirteen distinct groups, and most associated factors were in the same group. Multiple maternal factor groups were associated with the neonatal buccal, rectal, and stool microbiotas. Particularly, a higher maternal inflammatory state and a lower maternal socioeconomic position were associated with a higher alpha diversity of the neonatal buccal microbiota and beta diversity of the neonatal stool microbiota was influenced by maternal diet and cesarean section by 24–72 h postpartum. The risk of admission of a neonate to the newborn intensive care unit was associated with preterm birth as well as higher cytokine levels and probably higher alpha diversity of the maternal buccal microbiota.

At an early stage of life, gut and oral microbiotas seem to influence immune, metabolic, and other human developmental pathways[1–3]. A sub-optimal early-life gut microbiota has been associated with multiple adverse outcomes, including but not limited to obesity[4], diarrheal disease[5], Crohn's disease[6], type 1 diabetes[7], and necrotizing enterocolitis[8]. Thus, it is important to understand the mechanisms leading to the establishment of the early-life microbiota.

Immediately after birth (≤5 min), a vertical mother-neonate microbe transmission has been reported for vaginal deliveries[9,10]; i.e., the neonatal skin, oral and nasal microbiomes are similar in composition to the maternal vaginal microbiome[11]. In contrast, the microbiomes of cesarean section (C-section) neonates are more likely to resemble the maternal skin microbiome[11,12]. C-section impacts the neonatal nasal, oral, and skin microbiomes at an early-life stage[3,11,13–17], but this influence seems to be lost by six (6) weeks after birth[13]. In both vaginal and C-section deliveries, the abundance of the commonly

transmitted taxa seems to decrease rapidly with age[9,18]. These observations illustrate that the maternal microbiome seems only transiently to seed the neonatal microbiomes, and the impact of the maternal microbe seeding on the neonatal microbiomes diminishes rapidly after birth. By six weeks of age, the composition and function of the infant's skin, oral, nasal, and stool microbiotas have been expanded and diversified[13].

Within three days postpartum, the number of strictly anaerobic taxa in the oral and stool microbiotas is greatly decreased[9], suggesting neonatal environments experience greater exposure to oxygen than the maternal sites that initially contribute to the neonatal microbiota. In parallel, the relative abundance of facultative anaerobes in the stool microbiota increases while the relative abundance of aerobes decreases[9], which seems to be consistent with the observation of the transition from aerobic to anaerobic conditions in the gut[19–21]. The neonatal stool microbiota increases in richness immediately after

[1]Microbiology & Immunology, School of Medicine, Virginia Commonwealth University, Richmond, VA 23298, USA. [2]Center for Microbiome Engineering and Data Analysis, Virginia Commonwealth University, Richmond, VA 23298, USA. [3]Statistical Sciences and Operations Research, College of Humanities & Sciences, Virginia Commonwealth University, Richmond, VA 23284, USA. [4]Computer Science Department, College of Engineering, Virginia Commonwealth University, Richmond, VA 23298, USA. [5]Genomics Core, Virginia Commonwealth University, Richmond, VA 23298, USA. ✉e-mail: gregory.buck@vcuhealth.org

birth[11] and continues to increase throughout the first three years of life[22], indicating a continuous input of microbes from the environment.

Mode of delivery[3], breast or formula feeding[14,23], time in the newborn intensive care unit (NICU)[24], and other environmental factors have also been shown to impact the neonatal microbiotas. However, the mechanisms by which maternal factors impact the neonatal microbiotas remain obscure. Interestingly, transplantation of maternal fecal microbiota seeds stool microbiome to neonates born by C-section[25], suggesting that seeding from the maternal microbiome can contribute to the neonatal stool microbiome.

### Table 1 | Clinical and demographic characteristics of participants in the cohort

|  | Mothers |
|---|---|
| **Age** | |
| Mean (SD) | 27.0 (5.3) |
| Range | 15–40 |
| **Race / ethnicity** | |
| Black / African American | 111 (67.7) |
| White / Caucasian | 43 (26.2%) |
| Others / Unknown | 10 (6.1%) |
| **Preterm birth** | |
| Preterm | 28 (17.1%) |
| Term | 136 (82.9%) |
|  | **Babies** |
| **NICU admission** | |
| Yes | 17 (10.4%) |
| No | 139 (84.8) |
| Unknown | 8 (4.9%) |

More information is available in SI Data 1 sheet 2.

Previous studies have shown an association between maternal immune-related metabolites and the development of the fetal and neonatal immune system[16,26] and the interaction between the neonatal immune system and the neonatal microbiome[27]. Thus, maternal cytokines representing the immune state of the female reproductive tract could be potential factors influencing the neonatal microbiome.

Herein, we examine the maternal and neonatal microbiotas, the demographic and clinical metadata, and the maternal cytokine profiles of 164 mother-neonate dyads previously enrolled in the Multi Omic Microbiome Study Pregnancy Initiative[28,29]. The compositions of the neonatal oral, rectal, and stool microbiotas during the first three days postpartum are characterized, and models of how these changes are directed by maternal factors have been developed. Finally, maternal factors associated with the risk of admission of neonates into the NICU are characterized.

## Results

### Neonatal microbiotas within three days postpartum

The studied dataset was collected from 164 mother-neonate dyads. The data included: (i) 16S rRNA gene taxonomic profiles from neonatal buccal (NB), rectal (NR), and stool (NS) samples, and maternal buccal (MB), rectal (MR), and vaginal (MV) samples from the pregnant women; (ii) cytokine/chemokine expression levels in the vaginal fluid of the pregnant women; and (iii), maternal and neonatal clinical and related metadata (Table 1 and Supplementary Data 1). The experimental designs and case numbers are shown in Fig. S1 and Supplementary Data 1 sheet 2. Since many neonatal samples are missing in the cohort, a cross-sectional study without case match was performed to investigate the profiles of the neonatal microbiotas within three days postpartum (Fig. S1a).

Alpha diversities quantified by the Shannon index of the NB and NR microbiotas were lower on days 1 (24–48 h postpartum) and 2 (48–72 h postpartum) than on day 0 (0–24 h postpartum) (Fig. 1a; note, no day 0 stool samples were collected). There were no significant

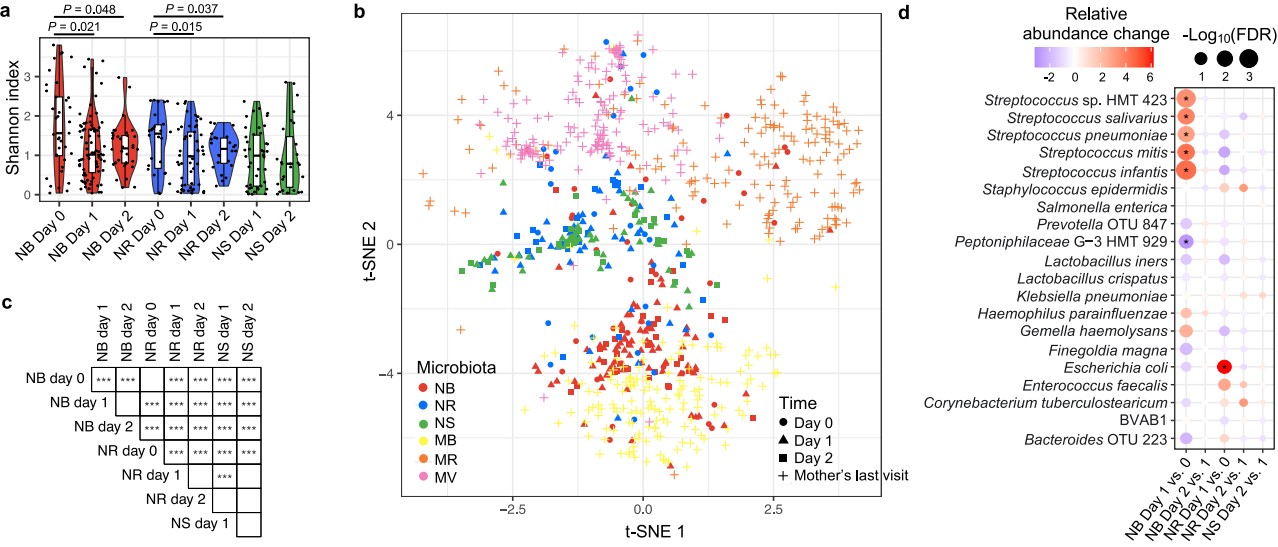

**Fig. 1 | The neonatal microbiotas within three days postpartum.** The neonatal buccal (NB), rectal (NR), and stool (NS) microbiotas on day 0 (within 24 h postpartum), day 1 (24–48 h postpartum), and day 2 (48–72 h postpartum) after birth and the maternal buccal (MB), rectal (MR), and vaginal (MV) microbiotas collected on the last visit of pregnancy were used in the diversity analyses. The experimental design, including case number, is shown in Fig. S1a. **a** Alpha diversity (Shannon index) of the neonatal microbiotas on different days. *P* values were quantified using the two-sided Mann–Whitney U test. Lines in the boxplots represent maximum, 75% quantile, median, 25 quantile, and minimum values from top to bottom. **b** Beta diversity of the NB, NR, and NS and MB, MR, and MV microbiotas quantified by the Bray-Curtis distance and visualized by the t-distributed stochastic neighbor embedding plot. **c** Pairwise analysis of the difference between two microbiotas was performed using the Adonis test with default parameters. The significance is indicated by asterisks. ***P value ≤ 0.001. **d** The significant changes in the relative abundance of taxa in the neonatal microbiotas are shown by a dot plot and are highlighted by asterisks. The relative abundance changes of taxa that are abundant in any studied microbiota are also visualized. Relative abundance change was tested by the ALDEx2 package with default parameters in R and quantified by the per-taxon median difference between two conditions. Adjusted *P* values were generated by the Benjamini-Hochberg correction of the Mann–Whitney U test.

differences between the alpha diversities of the NB, NR, or NS microbiotas from day 1 to day 2.

Beta diversity of the neonatal and maternal microbiotas was visualized in a t-distributed stochastic neighbor embedding (t-SNE) plot (Fig. 1b), and the differences among the neonatal microbiotas were determined by the Adonis test[30] (Fig. 1c). Similar to the alpha diversities reported above, the neonatal microbiotas at each body niche were significantly different in beta diversity between day 0 and later time points but not between days 1 and 2 (Fig. 1c). These results suggest that the NB and NR microbiotas change rapidly during days 0 and 1 but begin to stabilize by day 2. More specifically, the difference in the NB microbiota was caused by differences in both dispersions (Fig. S2a indicating variability among samples) and centroid (Fig. S2b representing the average characteristics of samples), but that in the NR microbiota was only caused by different dispersion. The within-group dissimilarities (an alternative way to examine dispersion) of both the NB and NR microbiotas decreased on days 1 and 2 (Fig. S2c). Thus, the composition of the NB microbiota obviously changed between day 0 and later time points because of significant differences in the centroid (Fig. S2b). Moreover, the microbiotas at each body site seem to converge among individuals after birth since the within-group dissimilarities were decreased (see more details in Supplementary Movie 1). Consistent with these observations, a significant difference in beta diversity between the NB and NR microbiotas was not observed on day 0 but appeared on days 1 and 2 (Fig. 1c and S2b).

Compared to the maternal microbiotas, the three neonatal microbiotas generally clustered more closely to each other on the t-SNE plot (Fig. 1b), suggesting a higher similarity among the neonatal microbiotas. Although our taxonomic profiles were largely limited to a species level, this observation is consistent with a shared source of taxa in these sampling sites at the beginning of life. Recent reports have suggested that the NB microbiota is similar to the MV microbiota immediately after birth (≤5 min)[11]. In the current study, on day 0 (≤ 24 h), the NB microbiotas were widely distributed on the t-SNE plot and some clustered among each of the three maternal microbiotas. These observations imply that, on the first day, microbes in the NB microbiotas in the different neonates may have derived from different maternal sources. Consistent with previous observations[23], the differences between the NB and MB microbiotas decreased with time, but the same phenomenon was not observed between the NR and MR microbiotas (Fig. S2d, e). These observations suggest that the composition of the NB microbiota tends to converge on the MB microbiota after childbirth, but the composition of the NR microbiota diverges from that of the MR microbiota at an early stage. However, the NB-MB distance in the paired mother-neonate dyads was not higher than in unpaired samples (Fig. S2f), indicating that general maternal factors shared by all the women were not the primary factors driving the composition of the NB microbiotas. The beta diversity of the NR microbiota on day 0 was most similar to that of the MV microbiota (Fig. 1b), suggesting that the main source of the initial NR microbiota may be the MV microbiota.

Consistent with beta diversity analysis, *Streptococcus* spp. were the predominant taxa in both the NB and MB microbiotas across all three time points (Fig. S2g). The predominant taxa in the NR microbiota on day 0 were similar to those abundant in the MV and MB microbiotas. However, the composition of the NR microbiota on days 1–2 differed from any of these three maternal microbiotas but was more similar to the NS microbiota; i.e., the dominant taxon was *Escherichia coli*.

Differential abundance analysis using 'ALDEx2'[31] showed that the relative abundance of the predominant taxa in both the NB and NR microbiotas on days 1 and 2, e.g., several *Streptococcus* spp. and *E. coli* respectively, increased from day 0 to day 1 (Fig. 1d). However, no significant changes were observed from day 1 to day 2. Together with observations on the alpha and beta diversity changes, these results suggested that the composition of the NB and NR microbiotas changed abruptly in the early hours after birth but were more stable by the third day postpartum.

## Association among maternal factors

Relationships among the maternal factors, i.e., the MB, MR, and MV microbiotas, cytokines in the maternal vaginal fluid, and maternal metadata, were investigated. The maternal samples and metadata were collected during the last pregnancy visit, most of which were in the third trimester (Fig. S3a). Although several maternal samples were missing, we found no significant differences among different types of maternal factors in data collection time (Fig. S3a). For simplification, the maternal microbiota data were converted to sets of values with a single dimension, i.e., Shannon index, evenness, and the number of observed taxa representing the alpha diversity and t-SNE1 and t-SNE2 (as shown in Fig. 1b) indicating the principal components. The microbiotas were classified into two groups, with the t-SNE values larger and lower than the mean of the t-SNE values, respectively. The biological meaning of the t-SNE values was illustrated by testing the abundance differences between the microbiotas with high and low levels of the t-SNE values (Fig. S3b–g). For example, Figure S3b shows the relative abundance of the taxa that are driving the first principle component of the MB microbiota in the t-SNE analysis. Figure S3b shows that *Streptococcus salivarius* is more abundant in the MB with a larger t-SNE1 value. Figure S3c–g shows similar analyses for the t-SNE values of the MB, MV, and MR microbiotas.

The maternal factor variables examined consist of binary, ordinal, or interval data. Thus, associations between data with different formats were tested by appropriate methods (see "Methods"). Significant associations among the maternal variables (Supplementary Data 2) were applied to establish a network map of maternal factors using the Gephi software[32]. Modularity analysis classified the maternal variables into thirteen modules (modules 0–12) (Fig. 2). These modules were named by centroid variables quantified by betweenness centrality or by the main components in each module, i.e., socioeconomic status, stress, smoking, disease, BMI, diet, sexual activity, MB, MR, MV, cytokine, preterm, and C-section modules. All the other variables were combined in module 13. The t-SNE1 of the MB microbiota was associated with smoking, and the MV microbiota was associated with the vaginal pH, a proinflammatory cytokine named IL-1β, and maternal factors in the socioeconomic status module. Additionally, many associations among the maternal metadata were discovered, some consistent with previous observations, e.g., associations between C-section and diabetes[33] and between preterm and several cytokines[28].

The 164 mother-neonate dyads were classified into two groups based on clustering the Gower's or Minkowski's distance (see "Methods") of variables in each module (Figs. 3a–d, 4b, d, 5a–b, and S4). Thus, all variables in the same module were converted to one variable with a value of 'A' or 'B'. The Chi-squared test illustrated no significant correlations among these modules (Supplementary Data 3). As a result, potential confounders or co-linear maternal factors were largely removed for the following analyses.

## Association between maternal factors and alpha diversity of neonatal microbiotas

Because the NB, NR, and NS microbiotas on day 2 were not significantly different from those on day 1 (Fig. 1), only one sample collected on day 1 or 2 was selected from each mother-neonate dyad in a case match design for the following association analyses. The maternal and neonatal samples were matched first by the same mother-neonate dyad and second by the closest distance in sample collection time. Thus, the maternal samples were collected on the last visit of pregnancy, and the neonatal samples were collected on the first visit on day 0 and the first sample on either day 1 or on day 2 if no sample was available on day 1 (Fig. S1b). The associations of maternal modules with the neonatal microbiotas on day 0 and those on day 1 or 2 were investigated, respectively.

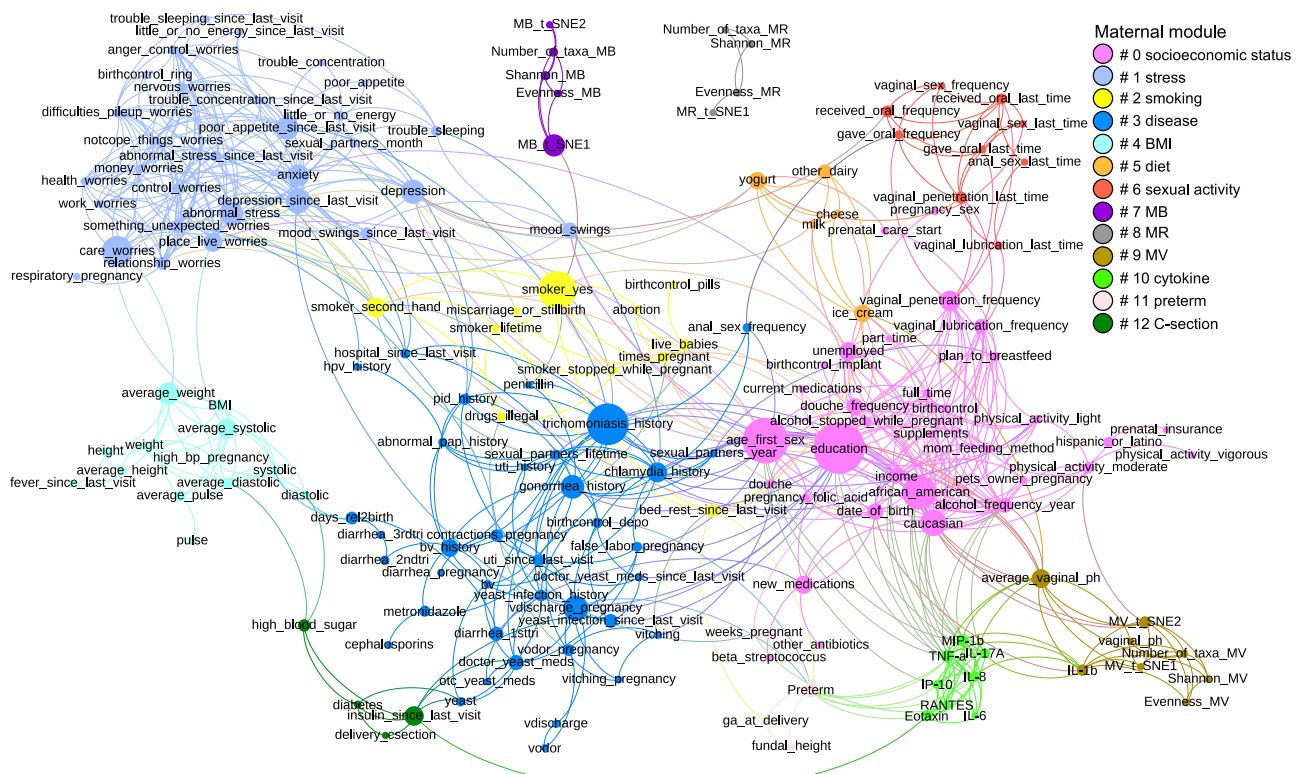

**Fig. 2 | Maternal factor network.** The associations between each pair of maternal variables were tested by different methods based on data types (see "Methods"). The significant associations were used as input to the Gephi software for the network analysis. The size of each node, representing each maternal factor, was determined by the betweenness centrality, and the modules were classified by the 'modularity' function in the Gephi software. The explanation of the maternal factors is shown in Supplementary Data 1, and the associations are listed in Supplementary Data 2.

Similar to the simplification of the maternal microbiotas described above, the neonatal microbiota data were converted to several variables with one dimension, i.e., Shannon index, evenness, the number of observed taxa, t-SNE1, and t-SNE2 (see the biological meaning of t-SNE values in Fig. S5). The association between the maternal modules and the neonatal variables described above was determined by the Mann–Whitney U test.

Cluster A in the socioeconomic status module, representing mothers with a high proportion of lower levels of education and income, African American, younger in the pregnancy, etc (Fig. 3a), and Cluster A in the diet module, indicating mothers consuming higher levels of yogurt, milk, cheese, ice cream, and other dairies (Fig. 3b), were associated with an increased Shannon index of the NB microbiota on day 1 or 2 (Fig. 3e, f). Additionally, higher levels of maternal vaginal proinflammatory cytokines (Fig. 3c) were associated with a larger number of observed taxa in the NB microbiota on day 1 or 2 (Fig. 3g). These results suggested that a higher level of alpha diversity in the NB microbiota on day 1 or 2 is a hallmark of a higher maternal inflammatory state and a lower maternal socioeconomic position.

A higher consumption of dairy (Fig. 3b) was associated with lower t-SNE2 values (Fig. 3h) or, in other words, more abundant *Streptococcus* spp. (Fig. 3i) in the NB microbiota on day 1 or 2. A higher frequency of sexual activity in the pregnant women was associated with a higher Shannon index of the NR microbiota on day 1 or 2 (Fig. 3j).

Interestingly, we observed no significant associations between the other maternal modules; e.g., modules 7, 8, and 9 representing the maternal buccal, rectal and vaginal microbiotas, and any of the neonatal microbiotas. Thus, a vertical microbe transmission from mother to neonate[9,10] contributes to the seeding and development of the neonatal microbiota, but our observations suggest that vertical transmission may not be the primary factor that determines the alpha diversity of the neonatal microbiotas within two days postpartum.

## Association between maternal factors and beta diversity of neonatal microbiotas

The influence of the maternal modules on the composition of the neonatal microbiotas was determined by the Adonis test using the marginal test model[30]. The results indicated that the composition of the NR microbiota on day 0 was influenced by the MV module (module 9), and that of the NS microbiota on day 1 or 2 was impacted by the diet and C-section modules (modules 5 and 12) (Fig. 4a). Distance-based redundancy analysis (dbRDA) showed that when the maternal vaginal microbiota had higher alpha diversity, higher vaginal pH, and more non-*Lactobacillus* spp. (cluster B in the MV module, Fig. 4b), e.g., *Gardnerella vaginalls* and bacterial vaginosis-associated bacterium 1 (BVAB1) (Fig. S3d, e), the NR microbiota on day 0 was composed of more taxa associated with dysbiosis of the maternal vaginal microbiota, e.g., *G. vaginalls* and BVAB1 (Fig. 4c). This observation implies a microbe transmission from the maternal vaginal microbiota to the neonatal gastrointestinal microbiota. Additionally, the NR samples on day 0 are obviously distributed in three subgroups on the RDA plot, implying multiple sources of microbes in the initiation of the NR microbiota. However, these phenomena disappeared after 24 h postpartum (Fig. 4a and Fig. S6). suggesting the influence of the MV module on the beta diversity of the NR microbiota was transient.

The dbRDA test also illustrated that if the mothers had lower levels of diet uptake (cluster B in the diet module, Fig. 3b) or the neonates were not delivered by C-section (cluster B in the C-section module, Fig. 4d), the NS microbiota on days 1 or 2 had more *E. coli* (Fig. 4e). Since *E. coli* is the most abundant and enriched taxon in the NS microbiota within three days postpartum (Fig. 1d and S2g), the

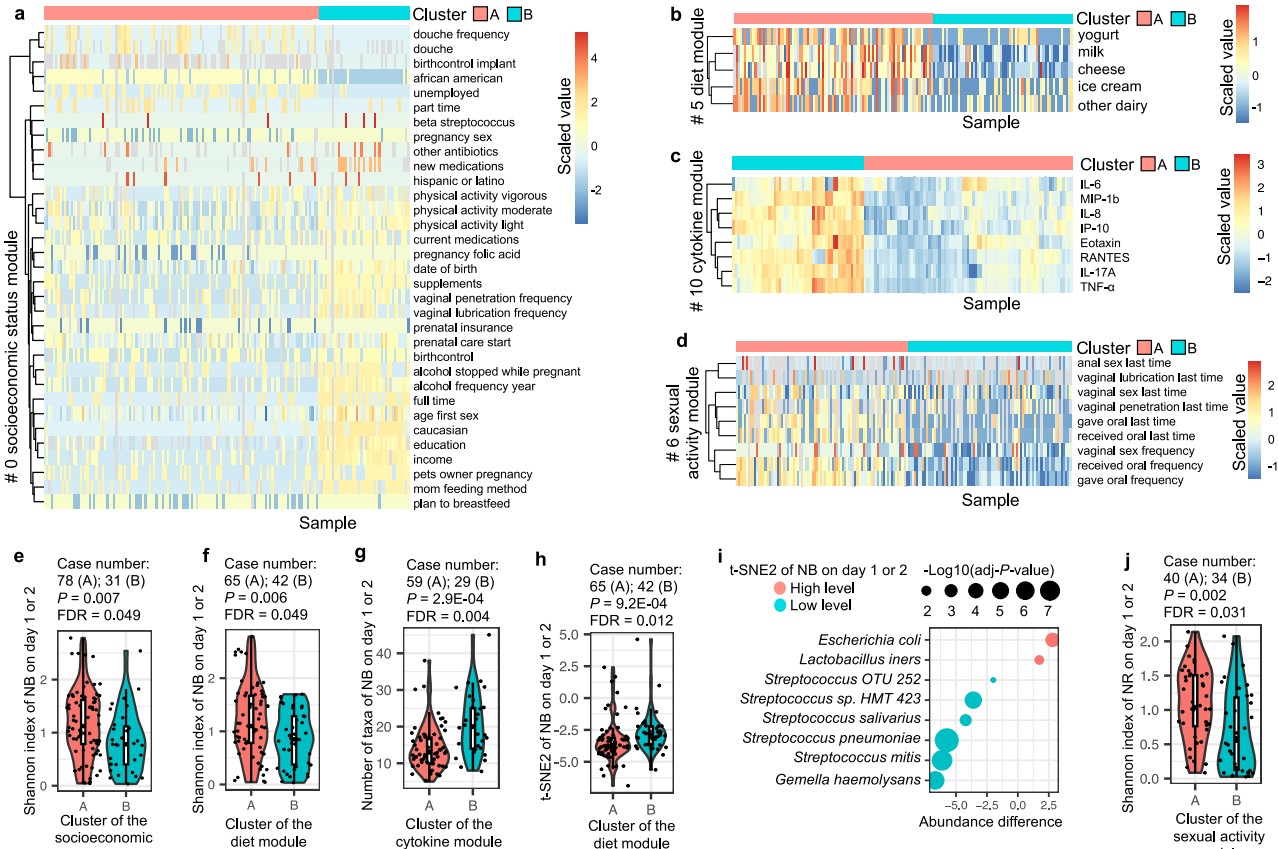

**Fig. 3 | Factors associated with the alpha diversity of the neonatal microbiotas.** The 164 maternal-neonatal dyads were classified into groups A and B according to the clustering of the Gower's or Minkowski's distance (see Methods) of maternal variables in each maternal factor module. The results of modules 0, 5, 10, and 6 are shown in (**a**–**d**). **e** The Shannon index of the NB microbiota on day 1 or 2 associated with the maternal factor module 0. **f** The Shannon index of the NB microbiota on day 1 or 2 associated with the maternal factor module 5. **g** The number of taxa of the NB microbiota on day 1 or 2 associated with the maternal factor module 10. **h** The t-SNE2 of the NB microbiota on day 1 or 2 associated with the maternal factor module 5. **i** The difference in the composition of the NB microbiota on day 1 or 2 between the t-SNE2 value at high and low levels. **j** The Shannon index of the NR microbiota on day 1 or 2 associated with the maternal factor module 6. Lines in the boxplots represent maximum, 75% quantile, median, 25 quantile, and minimum values from top to bottom. The two-sided Mann–Whitney U test was used to test the difference between alpha diversities of two microbiota groups with P values adjusted by the Benjamini-Hochberg procedure.

results suggest that vaginal delivery and a lower level of diet uptake of mothers are beneficial for a normal change of the NS microbiota during this time period.

## Maternal factors associated with the risk for NICU admission

NICU admission is an essential risk factor associated with mortality in both term and preterm birth neonates[34]. The NICU admission rates in the United States have risen from 6.62% in 2008 to 9.07% in 2018[35]. Here, odds ratio analysis indicated that preterm birth (module 11, Fig. 5a), higher levels of inflammatory cytokines (module 10, Fig. 3c), lower position of socioeconomic status (module 0, Fig. 3a), and higher alpha diversity of the MB microbiota (module 7, Fig. 5b) were associated with increased risks of NICU admission. However, only preterm birth and cytokine levels had significant associations with the NICU admission after correction for multiple testing (Fig. 5c), reflecting the fact that most babies born prematurely are admitted to the NICU[36]. Furthermore, similar odds ratio analysis was conducted among participants exhibiting specific delivery characteristics. Cytokine levels (module 10) were notably linked with NICU admission solely in term birth participants, rather than those born preterm (Fig. S7). These findings indicate an interaction between the influences of preterm birth and cytokines on the likelihood of NICU admission (Fig. S7b); however, cytokines can independently affect the risk of NICU admission among term birth participants (Fig. S7a).

The associations of these factors with the risks of NICU admission were also examined by establishing a machine-learning model using the random forest algorithm and a cross-validation strategy as previously described[37]. The area under the receiver operating characteristic (auROC), reflecting the quality of the model with a value between 0–1, is 0.777 (Fig. 5d). The importance of input variables was quantified by the mean decrease in Gini coefficient. Not surprisingly, gestational age at delivery, an interval value that reflects preterm birth, had the heaviest weight in predicting NICU admission. The Shannon index of the MB microbiota was the second most important variable. Additionally, parameters indicating sub-optimal maternal health conditions, e.g., expression of several pro-inflammatory cytokines, insufficient bed rest, yeast infection, hospitalization, etc., also contributed to the predictive model. The association among variables applied in the model was measured by Spearman's correlation. Most variables, e.g., inflammatory cytokine levels, bed rest, and douching, were correlated with preterm birth (Fig. 5e). Education was similar to preterm birth in correlation with cytokine levels, and yeast infection was correlated with education. The MB microbiota variables were independent of preterm birth and other variables.

In total, these results argue that both preterm birth and higher cytokine levels are risk factors for NICU admission. Moreover, the diversity of the MB microbiota could be a potential variable independent of preterm birth to increase or predict a higher risk of NICU admission.

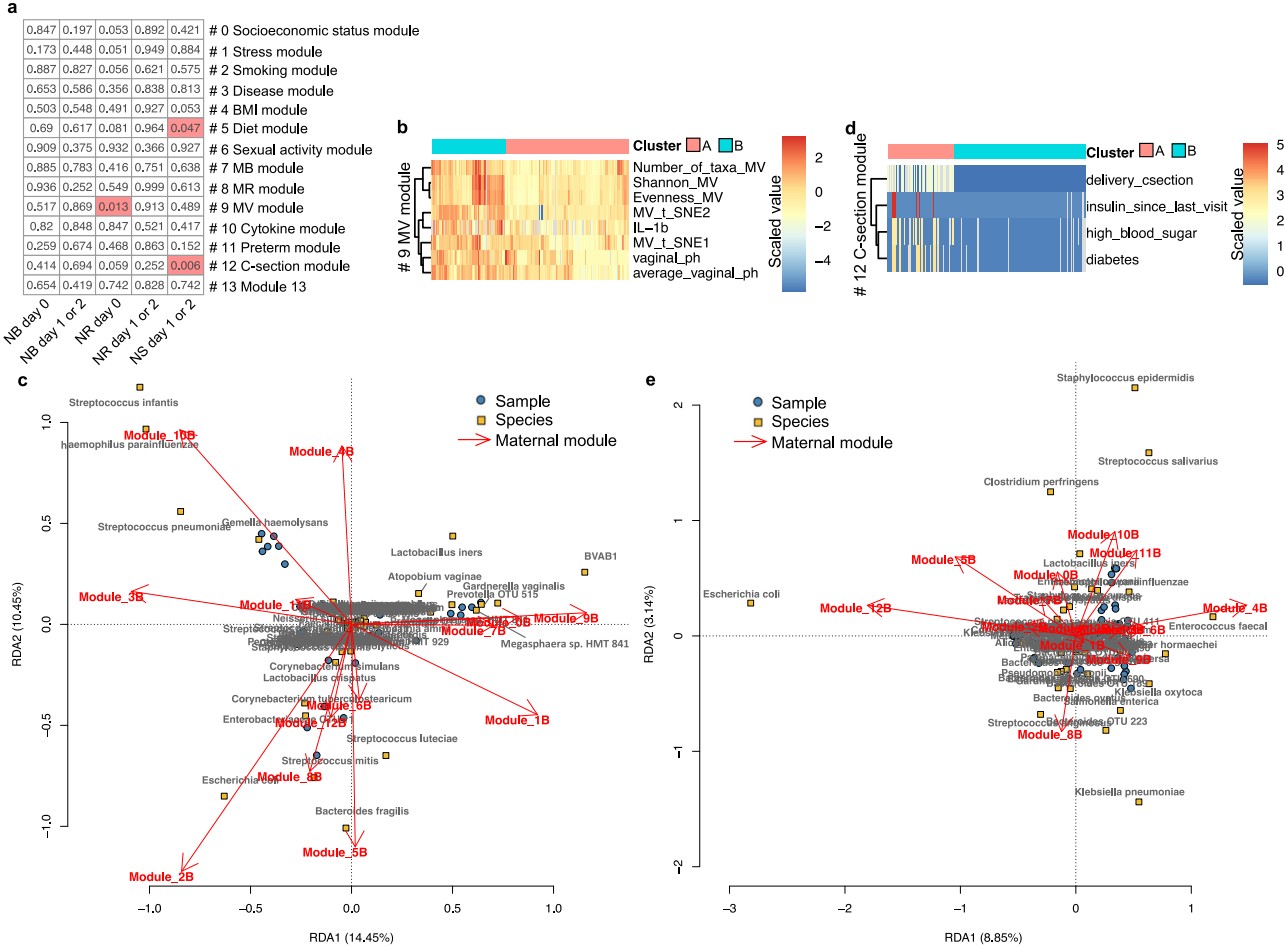

**Fig. 4 | Factors associated with the beta diversity of the neonatal microbiotas. a** The association between the maternal modules and the composition of the neonatal microbiotas was determined by the Adonis test with default parameters except for the parameter 'by' set as 'margin'. The Adonis *P* values are shown. **b** The classification of the 164 maternal-neonatal dyads into groups A and B according to the clustering of the Minkowski's distance of maternal variables in maternal factor module 9. **c** The dbRDA test to show the association between the maternal modules and the composition of the NR microbiota on day 0. A red arrow indicates the influence of a specific maternal module on the microbiota. For example, 'Module_9B' represents the enrichment of taxa driven by cluster B of module 9. **d** The classification of the 164 maternal-neonatal dyads into groups A and B according to the clustering of the Gower's distance of maternal variables in maternal factor module 12. **e** The dbRDA test to show the association between the maternal modules and the composition of the NS microbiota on day 1 or 2.

## Discussion

Our data indicate that the neonatal microbiotas, particularly the neonatal buccal microbiota, change rapidly to niche-specific profiles by 24–48 h after birth. Together with previous studies[22], the alpha diversity of the neonatal microbiota tends to decrease within two days postpartum and then continuously increase before age 3–5. The first stage of diversity change could be caused by the enrichment of specific microbes by local micro-environments in different body sites and the second stage could be driven from continuous input of microbes from environments.

Since the oral cavity is the gateway to the human body, local micro-environments in the oral cavity might be more easily affected by the external environment than those of the gastrointestinal tract, possibly explaining our observation of a higher similarity between the NB-MB microbiotas than that observed between the NR-MR microbiotas (Fig. S2d, e). Thus, oral taxa could be more easily transferred from mother to neonate by direct oral contact, breastfeeding, or oral contact with other fomites. Since anaerobes are abundant in newborns within one day postpartum[9], a higher level of oxygen in the oral cavity than in the gut could also lead to a faster change of the NB microbiota. However, the lack of similarity between the MR and the NR and NS microbiotas also reflects the fact that the neonatal gastrointestinal tract has a physiology different from that of mature women.

In a prior investigation utilizing a mouse model to explore the early development of small intestinal and colonic microbiotas, the authors observed a slight decrease in alpha diversity in both microbiotas from the initial time point (day 1) to the subsequent time point (day 7) and the compositions of the two microbiotas were similar on day 1 but became significantly different from day 7 onwards[38]. These findings align with our observations in the human NB and NR microbiotas (Fig. 1a, c). Previous studies have also demonstrated the transient transmission of maternal microbiome to the neonatal niche[9,18]. Therefore, it is plausible that the higher alpha diversity at the initial time point compared to the subsequent time point reflects the transient transfer of maternal microbes. Additionally, bacterial remnants from various sources to which a newborn is exposed upon amniotic membrane rupture could contribute to the higher alpha diversity at the initial time point.

Based on the high-quality metadata in the MOMS-PI project, we built a maternal factor network that linked most of the associated variables into the same module (Fig. 2). Thus, the influence of one module to the neonatal microbiota is largely independent of other modules (Supplementary Data 3). This network could also be applied in future studies to determine variables that must be matched in a case-matched design to remove potential confounders. Nevertheless, the absence of postpartum metadata, such as breastfeeding, restricted

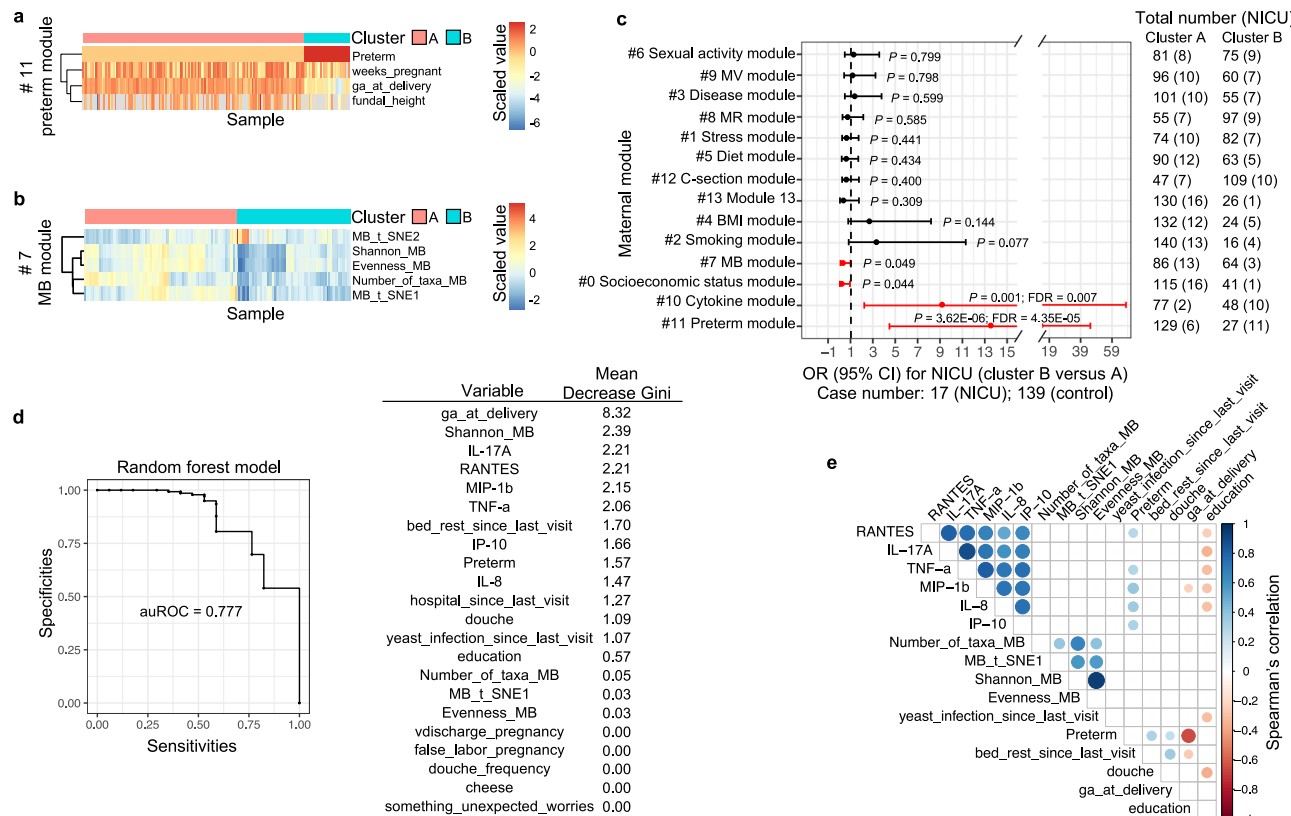

**Fig. 5 | Maternal factors associated with NICU admission.** The 164 maternal-neonatal dyads were classified to groups A and B according to the clustering of the Gower's distance of maternal variables in each maternal factor module. The results of modules 11 and 7 are shown on (**a**, **b**). **c** The association between the maternal modules and the risk of NICU admission was determined by the two-sided odds ratio analysis. Data are presented as risk ratios and upper and lower bounds of the estimate. The *P* values or *P* values with multiple testing corrections are shown. **d** A predictive model for the risk of the NICU admission was established using the random forest algorithm. The quality of the model, quantified by the area under the receiver operating characteristic (auROC), and the importance of input variables, determined by the mean decrease in Gini coefficient (Mean Decrease Gini), are shown. **e** The correlation among the input variables in the random forest model was tested by the Spearman's correlation. The correlation coefficients with significant correlations are shown by the heatmap.

this study from elucidating the association between the neonatal microbiota and post-delivery environmental factors.

Multiple maternal modules have been associated with the neonatal microbiotas in this study. However, the mechanisms responsible for these associations remain unknown. As described in the introduction, both the vertical maternal-neonatal microbe and metabolite transmissions could mediate the associations between maternal factors and the neonatal microbiota. We have tried to perform mediation analyses to investigate the roles of maternal cytokines and lipids in mediating the association between maternal factors and the neonatal microbiota (data not shown). For example, C20 ceramide seems to mediate the influence of maternal smoking and anxiety on the alpha diversity of the NB microbiota. However, largely due to a limited sample size, none of the mediation effects were significant after multiple testing corrections, and additional studies are warranted.

Previous studies have shown that *Fusobacterium nucleatum*, which can spread hematogenously from the oral cavity to the uterus and result in NICU admission[39], is more abundant in periodontitis, and periodontitis is associated with higher alpha diversity of the oral microbiota[40]. Thus, an MB microbiota with higher alpha diversity could include more opportunistic pathogens, e.g., *F. nucleatum*, and increase the risk of NICU admission. The association between the MB microbiota and NICU admission requires further studies with larger sample sizes.

## Methods
### Cohort
Data used in this study were produced under the umbrella of the Multi-Omic Microbiome Study-Pregnancy Initiative (MOMS-PI)

project[28,29], which enrolled ~1500 pregnant women with the goal of studying the contribution of the vaginal microbiota to adverse outcomes of pregnancy, including preterm birth. Here, we focused on the maturation of early-life neonatal microbiotas and variables influencing this process. The studied dataset was collected from 164 mother-neonate dyads, including 16 S rRNA taxonomic profiles from neonatal buccal, rectal, and stool sites, maternal buccal, rectal, and vaginal niches in the pregnant women, maternal and neonatal clinical and other metadata, and cytokine expression levels in the vaginal fluid of the pregnant women. Maternal samples and metadata were from the last pregnancy visit. Neonatal metadata was collected on the first visit after childbirth, and neonatal samples were collected on the first visit on day 0 (within 24 h postpartum), day 1 (24–48 h postpartum), and day 2 (48–72 h postpartum), but there were no neonatal stool samples from day 0. Metadata of the 164 mother-neonate dyads include the time of sample collection during or after pregnancy, gestational age at delivery, delivery method, maternal disease records, adverse outcomes of pregnancy, maternal stress level, body mass index, birth control methods, drug use, racioethnicity, diet, economic status, etc. (Supplementary Data 1).

### Statistics & reproducibility
The experimental design and case number for each analysis are shown in Fig. S1. The statistical methods are introduced in the other parts of the Methods as shown below. To reproduce the results, data and codes are shared as introduced in 'Data and code availability'. All the participants and samples available for this study are involved in this study.

The criteria for inclusion and exclusion of recruited participants are described in our previous study[28].

## Data processing

Raw 16 S rRNA sequencing data were treated by quality control, trimming, merging paired sequence reads, and removing human reads as previously described[28,41]. For better taxonomic profiling of the 16 S rRNA sequencing data to species level, a new 16 S rRNA V1-V3 region database was created based on the Greengenes database version gg_13_5 (https://greengenes.secondgenome.com/)[42] and the HOMD database version 15.1 (https://www.homd.org/)[43]. The 16 S rRNA sequences in the Greengenes and HOMD databases were mixed and sorted in the following order. The sequences with taxonomic annotations at the species level in the Greengenes database had the highest priority, followed by all the sequences in the HOMD database, the Greengenes sequences with annotations at the genus level, and then the Greengenes sequences with annotations at levels higher than genus level. The V1-V3 region of the full-length 16 S rRNA sequences was extracted using the V-Xtractor[44]. Finally, V1-V3 sequences in the database with a similarity higher than 97% are filtered using USEARCH[45] so that only one remains in the database. The trimmed, merged, and filtered 16 S rRNA raw data were assigned to the species level using the new 16 S rRNA V1-V3 region database (https://github.com/GregoryBucklab/Neonatal_microbiome_project/) to generate the feature tables of the microbiotas. The MV microbiotas have been sequenced by metagenomic sequencing in the MOMS-PI project, and reads of taxa in the 16 S rRNA feature table of the MV microbiotas are all significantly, strongly, and linearly correlated with the reads of matched taxa in the metagenomic sequencing feature table (data not shown). Pretreated 16 S rRNA sequencing data were aligned to the new database for taxonomic assignment. An alignment with a similarity lower than 97% was assigned as 'BT' (below the threshold). Rarefaction curves indicate that the number of observed taxa in the studied microbiotas remains stable once the sequencing depth reaches 5000 (Fig. S9). Thus, samples with less than 5,000 total reads in the 16 S rRNA feature tables were eliminated from this study. Additionally, the 16 S rRNA feature tables were pre-filtered by a taxon threshold in which only taxa with relative abundances higher than 0.1% in more than 5% of samples or no less than 1 read in more than 15% of samples were kept (Supplementary Data 4).

## Blank controls

Water controls were collected and sequenced with other samples in every sequencing in the MOMS-PI project. Similar to the previously reported[46], most of the 155 control samples have very low sequencing reads (Fig. S8a), and only limited taxa have high levels of read counts (Fig. S8b). The 16 S rRNA profiles of the studied microbiotas were filtered by total sample read and taxon read thresholds as introduced above. After the pre-treatment, the studied 16 S rRNA reads tables only contained high-quality samples and abundant or frequently detected taxa. The reads tables for the NB, NR, and NS microbiotas had 78, 72, and 75 taxa, respectively. As shown in Supplementary Data 5, only one taxon named *Fusobacterium nucleatum subsp. animalis* had an average reads count close to that detected in the neonatal buccal samples. Thus, this taxon was removed in this study. Most studied taxa had over 100-fold average reads in the neonatal samples compared with that in the water controls. A few taxa, i.e., *Prevotella bivia, Gardnerella vaginalis, Afipia broomeae,* and *Propionibacterium acnes*, had 10–100 fold average reads in the neonatal samples compared with that in the water controls, but none of these taxa were abundant in any of the studied neonatal microbiotas. Furthermore, six samples from each type of neonatal microbiota, along with six water controls, were randomly chosen. q-PCR was then conducted with two duplicates for each sample to measure bacterial biomass using primers for 16 S rRNA gene amplification (5'-CCTACGGGDGGCWGCA-3', 5'-GGACTACHVGGGTMT

CTAATC-3')[47]. The standard curve was prepared using a plasmid containing 16 S rRNA gene of *G. vaginalis* as the template for the q-PCR assay. The bacterial biomass of the NB, NR, and NS microbiotas was significantly higher compared to the blank controls (Fig. S8c).

## Diversity analysis

The 16 S rRNA taxonomic feature tables were normalized by rarefaction to the depth of the lowest number of reads in the samples (5,000) for diversity analysis. Alpha diversity, quantified by the Shannon index, evenness, and the number of observed taxa, were evaluated using the 'vegan' package in R[30]. The two-sided Mann–Whitney U test was used to test the difference between alpha diversities of two microbiotas. Beta diversity was measured and visualized by a t-distributed stochastic neighbor embedding (t-SNE) of Bray-Curtis distances using the 'Rtsne' package in R[48]. The difference in beta diversity between two microbiotas quantified by the Bray-Curtis distance was measured by a PERMANOVA analysis using the 'adonis2' function in the 'vegan' package[30]. The change of the Bray-Curtis distance between two microbiotas with the time after birth was quantified by the two-sided Kruskal–Wallis test. The difference in sample dispersion in the PERMANOVA analysis was quantified by the PERMDISP test using the 'betadisper' and 'adonis2' functions in R. The difference between within-group Bray-Curtis distances of two microbiotas was measured by the multiple response permutation procedure (MRPP) test using the 'mrpp' function in the 'vegan' package in R.

## Change of relative taxon abundance

The same taxon threshold was applied to pretreat the 16 S rRNA taxonomic feature tables introduced in the diversity analysis. Differential abundance analysis was performed using the 'ALDEx2' package in R[31]. The adjusted *P* value of relative abundance differences was tested by the 'aldex.ttest' function[31] using the two-sided Mann-Whitney U test value, followed by the Benjamini-Hochberg correction. The relative abundance change was measured by the 'aldex.effect' function and quantified by the per-taxon median difference between two conditions.

## Association among maternal factors

There are three data types in the maternal variable table, i.e., binary, ordinal, and interval data (Supplementary Data 1). The association between interval and ordinal data or among interval or ordinal data was measured by Spearman's correlation. The association of binary data with interval or ordinal data was tested by the Point-Biserial Correlation. The association among binary data was determined by the chi-squared test. *P* values in the three association analyses were adjusted by the Benjamini-Hochberg procedure, respectively.

## Maternal factor network

Each maternal factor dyad with two factors significantly associated with each other (adjusted *P* values less than or equal to 0.05) was input to the Gephi software as an edge to build an unweighted network[32]. All the following analyses were performed using the Gephi software. The structure of the network was built by the Force Atlas algorithm. The modules were distinguished by the 'Modularity' function with a resolution equal to 0.7. The betweenness centrality was calculated by the 'Avg. Path Length' function. The color and size of each factor node were determined by the module and betweenness centrality, respectively.

## Classification of maternal-neonatal dyads by maternal factors in each module

The Gower's distance among maternal-neonatal dyads was calculated by the 'gower.dist' function in R based on all maternal variables in the same module. The Gower's distance can calculate the distance of data with both categorical and interval values. However, when all maternal variables in the same module were interval data, i.e., those in modules 7–10, Minkowski's distance was used to calculate the distance among

maternal-neonatal dyads by the 'dist' function in R. The maternal-neonatal dyads were clustered according to their Gower's or Minkowski's distance using the 'ward.D' method in R. To simplify the interpretation of outcomes, the maternal-neonatal dyads were classified into two groups, i.e., group 'A' and 'B', using the 'cutree' function in R. To facilitate the visualization of the characteristics of maternal factors in the same module, maternal factors with binary and ordinal data were converted to numeric data using the 'xtfrm' function in R, and all the maternal factors in the same maternal module were normalized by the 'scale' function in R to generate the heatmaps.

### Association between maternal modules and neonatal microbiotas

To simplify the neonatal microbiota data, they were converted to a set of interval values with one dimension, i.e., Shannon index, evenness, and the number of observed taxa representing the alpha diversity and t-SNE1 and t-SNE2 (as shown in Fig. S5) indicating the main components. Since the maternal-neonatal dyads were classified into two groups, i.e., 'A' and 'B', the association between one maternal module and one neonatal microbiota variable was determined by the Mann-Whitney U test. P values in tests with the same dependent variable, or one-dimensional neonatal variable, were adjusted by the Benjamini-Hochberg procedure.

### Association between maternal modules and the composition of the neonatal microbiotas

The association between the maternal modules, with categorical values 'A' and 'B', and the composition of the neonatal microbiota was determined by the Adonis test with the parameter 'by' equal to "margin" and the dbRDA test using the 'capscale' function with the parameter 'distance' equal to "bray" in R[30].

### Association between maternal modules and the risk of NICU admission

The association between maternal modules, with categorical values 'A' and 'B', and the risk of NICU admission was determined by the odds ratio analysis using the ' oddsratio' function in R. The P values were adjusted by the Benjamini-Hochberg procedure.

### Predictive modeling of the risk of NICU admission using maternal factors

Ordinal data was converted to binary data by classifying each set of ordinal data into two groups, i.e., levels high and low, with the same or similar sample sizes. The binary data were then converted to numeric data using the one-hot encoding. The association between each maternal variable and NICU admission was tested by Spearman's correlation. Only the maternal variables with significant associations with NICU admission (without multiple testing corrections) were selected as input in the following machine learning model. The selected maternal variables are shown in Fig. 5d, and their correlations were tested by Spearman's correlation with P values adjusted by the Benjamini-Hochberg procedure. Missing values were imputed using the 'rfImpute' function in R[49]. A random forest algorithm and the 'randomForest' package in R[50] were applied to create the predictive model as previously described[37] using the selected maternal factors as independent variables and the outcome of the NICU admission as a dependent variable. The quality of the model was measured by the area under the receiver operating characteristic curve (auROC), and the importance of variables in the model was measured by the mean decrease in Gini coefficient.

### Resource and tools availability

Further information and requests for resources should be directed to and will be fulfilled by the lead contact, Gregory A. Buck (gregory.buck@vcuhealth.org). All the tools used in this study are shown in Supplementary Table 1. All Supplementary datasets and video are available on Figshare (https://doi.org/10.6084/m9.figshare.21905397).

### Reporting summary

Further information on research design is available in the Nature Portfolio Reporting Summary linked to this article.

## Data availability

Raw 16S rRNA sequences, cytokine data, and limited metadata of the Multi-Omic Microbiome Study-Pregnancy Initiative (MOMS-PI) project[28,29] have been deposited in the HMP DACC. Controlled-access data for all subjects in the MOMS-PI project have been deposited at the National Center for Biotechnology Information's controlled-access dbGaP (study no. 20280; accession ID phs001523.v1.p1; https://www.ncbi.nlm.nih.gov/projects/gap/cgi-bin/study.cgi?study_id=phs001523.v1.p1) and the SRA under BioProject IDs PRJNA326441, PRJNA326442, and PRJNA326441. Other researchers can reproduce the reported analysis using the datasets listed above.

## Code availability

All the codes have been deposited on GitHub (https://github.com/GregoryBucklab/Neonatal_microbiome_project) with a https://doi.org/10.5281/zenodo.11200086.

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

## Acknowledgements

We gratefully acknowledge the women and neonates who provided samples for these analyses. We also acknowledge other members of the Vaginal Human Microbiome Consortium at VCU who participated in the collection, processing, and analysis of samples from the women and neonates who participated in this project. This work was funded by grants UH3AI083263 (G. A. B.), U54HD080784 (G. A. B.), and R01HD092415 (G. A. B.) from the National Institutes of Health and the GAPPS BMGF PPB grant from the Global Alliance to Prevent Prematurity and Stillbirth (G. A. B.).

## Author contributions

G. A. B., M. G. S., and B. Z. conceived and designed the study. B. Z. performed the analyses with the assistance of D. J. E. and wrote the paper. K. M. S. and A. M. participated in sample collection and sequencing. L. E. performed pretreatment of the raw sequencing data. G.A.B., D. J. E., and M.G.S. edited the paper.

## Competing interests

G.A.B. is a member of the Scientific Advisory Board of Juno, LTD.; a startup biotech firm focused on using the vaginal microbiome to address issues of women's gynecologic and reproductive health. The remaining authors declare no competing interests.

**Institutional review board statement and informed consent statement**
Participants were enrolled in the Multi-Omic Microbiome Study: Pregnancy Initiative (MOMS-PI) and the Vaginal Human Microbiome Project under the umbrella of the National Institutes of Health Human Microbiome Project (https://commonfund.nih.gov/hmp). Women were enrolled in women's clinics associated with the Virginia Commonwealth University Health Center. Study protocols were approved by the Virginia Commonwealth University institutional review board under protocols

IRB# HM12169 or HM15527. Written informed consent or parental permission and assent were provided by participants or minors older than 15 years, respectively. Exclusion criteria included women incapable of understanding the informed consent or assent forms or who were incarcerated. Demographic, health histories, dietary assessments, and clinical data (e.g., gestational age, height, weight, blood pressure, vaginal pH, diagnosis, etc.) were collected. Clinical information about neonates was collected at birth (day 0) and at 24–48 h (day 1) or 48–72 h (day 2) after birth. Other exclusion criteria included: (1) inability to self-sample due to any reason; (2) significant vaginal bleeding; (3) ruptured membranes; (4) herpes lesions.
