## [Peer Review File · Nature Communications]

REVIEWER COMMENTS

Reviewer #1 (Remarks to the Author):

The authors have worked hard and significantly improved the robustness of the analysis and reliability of the reporting. For the most part, my comments have been addressed. However, it is disappointing that basic statistical practice (e.g., adjusting for confounders and repeated measures) was not considered in the first submission. As a result, the updated and more statistically meaningful analysis has largely disproved the most exciting and novel results from the initial submission (e.g., maternal lipids and vertical transmission).

Reviewer #1 (Remarks on code availability):

The authors should be commended for sharing the code

Reviewer #2 (Remarks to the Author):

Authors covered most of my questions, however, still some points remain to be modified.

- Please, do not include "microbiome" as just 16S data is reported. "Microbiota" would be the word.

-Blanks have been included but this does not report information about the quantification of microbial biomass.

- Figure 1d, please, microbial genus in italics

-5000 reads for rarefaction is too low. Not sure if with these coverage the plateau is reached. Please, provide more info.

- The risk of NICU admission would be higher in preterm. Did the authors checked for both populations independently? Preterm (n=28) has higher risk of NICU. Please, provide more details.
- A clinical table of mother-infants with basic characteristics has to be included in the main manuscript.

Reviewer #2 (Remarks on code availability):

Code is appropriate

Reviewer #3 (Remarks to the Author):

The authors have made substantial revisions and redesigned their manuscript in which the entire focus shifted from maternal transmission to the comparisons of microbiota composition across body sites and time and the impact of maternal factors on the infant microbiota of the skin, rectal swabs and stool samples. Most of my previous concerns, including the need for metagenomics data to draw conclusions on transmission and the inadequate study design and methods for their previous primary study aims have thereby been adequately addressed. By focussing on the microbiota composition across body sites and throughout the first days of life and the link to maternal factors the manuscript is now also more noteworthy since few prior studies have examined this in such depth and scale, particularly for the first few days of life.

I only have a few remaining comments:

- Lines 43-46 in the introduction appear confusing to me. At first the authors state that the reduction in strict anaerobes in the neonatal gut during the first days could be the result of increased oxygen exposure as compared to the adult gut, which seems logical. However in the following sentence the authors state that the concomitant increase in facultative anaerobes could be the result from the transition from an aerobic to anaerobic environment. The latter seems counterintuitive – do the authors mean a transition from an anaerobic to aerobic environment or could they specify what their reasoning is?

- The authors show a decrease in alpha diversity in NB and NR from day 0 to day 1-2. Moreover they show that on day 0 there is no significant difference between NB and NR while communities become significantly different from day 1 onwards. The authors might want to compare this with the dynamics in microbiota establishment over the first hours and days in some of the published studies

in animals (e.g., doi: 10.1038/s41467-020-17183-8), which are in line with these findings. Also some more discussion on this effect could be included in the discussion section. For example, might the high alpha diversity in the first hours/day just be a reflection of the transient passage of not only bacteria but also bacterial remnants from all kinds of sources a newborn is exposed to upon rupture of the amniotic membranes?

- The authors state 16S rRNA profiling throughout their manuscript, but as far as I can check they performed DNA profiling rather than transcriptomics. Please revise to 16S rRNA gene profiling or sequencing of the 16S rRNA gene.

- Could the authors explain why numbers of neonatal samples in the matched design (FigS1b) in some cases exceed the number of neonatal samples in the cross-sectional design (Fig1a). I would expect unpaired samples to be as high or even higher than when samples are paired with maternal samples.

For example NB at day 1+ 2 are 110 (Fig1a), but 112 in the matched design. For NS at day 1+2 are 66 vs 69 in the matched design.

- Please clarify/extend the statement regarding dispersion on lines 79-81, e.g. how does this affect your conclusion on the effect of NB and NR microbiota in the lines below (82-86) ?

- Please address in the manuscript that the lack of information on breastfeeding is a real limitation

- In lines 171-172 it is stated that: "but our observations suggest that vertical transmission may not be the primary factor that determines the 172 alpha diversity of the neonatal microbiotas". Authors should emphasize herein that the observations only entail short-term effects (day 0-2).

Reviewer #1 (Remarks to the Author):

The authors have worked hard and significantly improved the robustness of the analysis and reliability of the reporting. For the most part, my comments have been addressed. However, it is disappointing that basic statistical practice (e.g., adjusting for confounders and repeated measures) was not considered in the first submission. As a result, the updated and more statistically meaningful analysis has largely disproved the most exciting and novel results from the initial submission (e.g., maternal lipids and vertical transmission).

Response: We acknowledge the reviewer's comments, but due to insufficient lipid data, we cannot draw robust conclusions. Therefore, we have opted not to include them in this manuscript.

Reviewer #2 (Remarks to the Author):

- Please, do not include "microbiome" as just 16S data is reported. "Microbiota" would be the word.

Response: Thank you for the reminder. In the updated version, we have substituted "microbiome" with "microbiota".

-Blanks have been included but this does not report information about the quantification of microbial biomass.

Response: The results of the microbial biomass has been added at lines 315-319 and in Fig. S8c.

- Figure 1d, please, microbial genus in italics

Response: Microbial names in Figure 1d have been corrected. Thanks.

-5000 reads for rarefaction is too low. Not sure if with these coverage the plateau is reached. Please, provide more info.

Response: Thanks for the reminder. We have added Fig. S9 for rarefaction curve and related text at line 298. "Rarefaction curves indicate that the number of observed taxa in the studied microbiotas remains stable once the sequencing depth reaches 5000 (Fig. S9)."

- The risk of NICU admission would be higher in preterm. Did the authors checked for both populations independently? Preterm (n=28) has higher risk of NICU. Please, provide more details.

Response: Thanks for the comment. We have added a new Fig. S7 and related text at lines 200-204. 'Furthermore, similar odds ratio analysis was conducted among participants exhibiting specific delivery characteristics. Cytokine levels (module 10) were notably linked with NICU admission solely in term birth participants, rather than those born preterm (Fig. S7). These findings indicate an interaction between the influences of preterm birth and cytokines on the likelihood of NICU admission (Fig. S7b); however, cytokines can independently affect the risk of NICU admission among term birth participants (Fig. S7a).'

- A clinical table of mother-infants with basic characteristics has to be included in the main manuscript.

Response: We added Table 1 to show the basic characteristics. More information is provided in SI Data 1 sheet 2.

Reviewer #3 (Remarks to the Author):

- Lines 43-46 in the introduction appear confusing to me. At first the authors state that the reduction in strict anaerobes in the neonatal gut during the first days could be the result of increased oxygen exposure as compared to the adult gut, which seems logical. However in the following sentence the authors state that the concomitant increase in facultative anaerobes could be the result from the transition from an aerobic to anaerobic environment. The latter seems counterintuitive – do the authors mean a transition from an anaerobic to aerobic environment or could they specify what their reasoning is?

Response: Thanks for pointing out the confusing part. It is possible that bacterial metabolism can consume oxygen and convert the gut environment from aerobic to anaerobic. However, we would not like to state any causal relationship. Thus, to make to more clear, the sentence is modified at line 45. 'In contrast, the relative abundance of facultative anaerobes in the stool microbiota increases but that of aerobes decreases⁹, which seems to be consistent with the transition from aerobic to anaerobic conditions in the gut¹⁹⁻²¹.'

- The authors show a decrease in alpha diversity in NB and NR from day 0 to day 1-2. Moreover they show that on day 0 there is no significant difference between NB and NR while communities become significantly different from day 1 onwards. The authors might want to compare this with the dynamics in microbiota establishment over the first hours and days in some of the published studies in animals (e.g., doi: 10.1038/s41467-020-17183-8), which are in line with these findings. Also some more discussion on this effect could be included in the discussion section. For example, might the high alpha diversity in the first hours/day just be a reflection of the transient passage of not only bacteria but also bacterial remnants from all kinds of sources a newborn is exposed to upon rupture of the amniotic membranes?

Response: Great idea, and thanks for the information on the publication. We have added the discussion and cited the mentioned paper in lines 235-243. 'In a prior investigation utilizing a mouse model to explore the early development of small intestinal and colonic microbiotas, the authors observed a slight decrease in alpha diversity in both microbiotas from the initial time point (day 1) to the subsequent time point (day 7) and the compositions of the two microbiotas were similar on day 1 but became significantly different from day 7 onwards. These findings align with our observations in the human NB and NR microbiotas (Fig. 1a and c). Previous studies have also demonstrated the transient transmission of maternal microbiome to neonatal microbiome¹⁶. Therefore, it is plausible that the higher alpha diversity at the initial time point compared to the subsequent time point reflects the transient transfer of maternal microbes. Additionally, bacterial remnants from various sources to which a newborn is exposed upon amniotic membrane rupture could contribute to the higher alpha diversity at the initial time point.'

- The authors state 16S rRNA profiling throughout their manuscript, but as far as I can check they performed DNA profiling rather than transcriptomics. Please revise to 16S rRNA gene profiling or sequencing of the 16S rRNA gene.

Response: Thanks for the reminder. We have checked to make sure the data are claimed as 16S rRNA gene profiling in the manuscript.

- Could the authors explain why numbers of neonatal samples in the matched design (FigS1b) in some cases exceed the number of neonatal samples in the cross-sectional design (Fig1a). I would expect unpaired samples to be as high or even higher than when samples are paired with maternal samples. For example NB at day 1+ 2 are 110

(Fig1a), but 112 in the matched design. For NS at day 1+2 are 66 vs 69 in the matched design.

Response: Thank you for bringing the mistake to our attention. The numbers in Fig. S1 were derived from an outdated version employing distinct exclusion strategies. We have rectified this in the updated Fig. S1a.

- Please clarify/extend the statement regarding dispersion on lines 79-81, e.g. how does this affect your conclusion on the effect of NB and NR microbiota in the lines below (82-86) ?

Response: Thanks for figuring out the confusing part. We have rewritten this paragraph in lines 77-90, as shown below. 'Beta diversity of the neonatal and maternal microbiotas was visualized in a t-distributed stochastic neighbor embedding (t-SNE) plot (Fig. 1b), and the differences among the neonatal microbiotas were determined by the Adonis test[®] (Fig. 1c). Similar to the alpha diversities reported above, the neonatal microbiotas at each body niche were significantly different in beta diversity between day 0 and later time points but not between days 1 and 2 (Fig. 1c). These results suggest that the NB and NR microbiotas change rapidly during days 0 and 1 but begin to stabilize by day 2. More specifically, the difference in the NB microbiota was caused by differences in both dispersion (Fig. S2a indicating variability among samples) and centroid (Fig. S2b representing the average characteristics of samples), but that in the NR microbiota was only caused by different dispersion. The within-group dissimilarities (an alternative way to examine dispersion) of both the NB and NR microbiotas decreased on days 1 and 2 (Fig. S2c). Thus, the composition of the NB microbiota obviously changed between day 0 and later time points because of significant differences in the centroid (Fig. S2b). Moreover, the microbiotas at each body site seem to converge among individuals after birth since the within-group dissimilarities were decreased (see more details in Supplementary Movie 1). Consistent with these observations, a significant difference in beta diversity between the NB and NR microbiotas was not observed on day 0 but appeared on days 1 and 2 (Fig. 1c and S2b).'

- Please address in the manuscript that the lack of information on breastfeeding is a real limitation

Response: Thanks for the comment. One sentence is added in Discussion at line 248.

'Nevertheless, the absence of postpartum metadata, such as breastfeeding, restricted this study from elucidating the association between the neonatal microbiota and post-delivery environmental factors.'

- In lines 171-172 it is stated that: "but our observations suggest that vertical transmission may not be the primary factor that determines the alpha diversity of the neonatal microbiotas". Authors should emphasize herein that the observations only entail short-term effects (day 0-2).

Response: Agree. The sentence is modified as 'but our observations suggest that vertical transmission may not be the primary factor that determines the alpha diversity of the neonatal microbiotas within two days postpartum.' at line 173.